# Climate Change Effects on the Predicted Heat Strain and Labour Capacity of Outdoor Workers in Australia

**DOI:** 10.3390/ijerph20095675

**Published:** 2023-04-28

**Authors:** Andrew P. Hunt, Matt Brearley, Andrew Hall, Rodney Pope

**Affiliations:** 1School of Biomedical Sciences, Faculty of Health, Queensland University of Technology (QUT), Brisbane, QLD 4059, Australia; 2Thermal Hyperformance, Hervey Bay, QLD 4655, Australia; 3National Critical Care and Trauma Response Centre, Darwin, NT 0800, Australia; 4School of Allied Health, Exercise & Sports Sciences, Charles Sturt University, Albury, NSW 2640, Australia; 5Gulbali Institute, Charles Sturt University, Albury, NSW 2640, Australia; 6Tactical Research Unit, Bond University, Robina, QLD 4229, Australia

**Keywords:** predicted heat strain, climate change, wet-bulb globe temperature, health and safety, labour capacity, workability, dehydration, predicted sweat loss

## Abstract

Global heating is subjecting more of the planet to longer periods of higher heat stress categories commonly employed to determine safe work durations. This study compared predicted worker heat strain and labour capacity for a recent normal climate (1986–2005) and under commonly applied climate scenarios for the 2041–2080 period for selected Australian locations. Recently published heat indices for northern (Darwin, Townsville, and Tom Price) and south-eastern coastal and inland Australia locations (Griffith, Port Macquarie, and Clare) under four projected climate scenarios, comprising two representative concentration pathways (RCPs), RCP4.5 and RCP8.5, and two time periods, 2041–2060 and 2061–2080, were used. Safe work durations, before the threshold for core temperature (38.0 °C) or sweat loss (5% body mass) are attained, were then estimated for each scenario using the predicted heat strain model (ISO7933). The modelled time to threshold core temperature varied with location, climate scenario, and metabolic rate. Relative to the baseline (1986–2005), safe work durations (labour capacity) were reduced by >50% in Port Macquarie and Griffith and by 20–50% in northern Australia. Reaching the sweat loss limit restricted safe work durations in Clare and Griffith. Projected future climatic conditions will adversely impact the predicted heat strain and labour capacity of outdoor workers in Australia. Risk management strategies must adapt to warming conditions to protect outdoor workers from the deleterious effects of heat.

## 1. Introduction

Global heating is exposing outdoor workers to increased risk of occupational heat strain and consequent adverse impacts on health and labour capacity [1,2]. Rising seasonal heat and the frequency of extreme heat events are placing outdoor workers at risk of heat-related illness and mortality in more locations for longer periods of the year [3]. Recent modelling suggests that, on average, wet-bulb temperatures will rise by 2–3 °C in the 21st century in the tropics and mid-latitudes [4]. As wet-bulb temperature rises, the body’s principal avenue for heat loss, the evaporation of sweat, becomes less efficient. Workers face either reduced productivity or an increased risk of exertional heat illness. Indeed, in 2021, heat exposure led to the loss of an estimated 470 billion labour hours worldwide, which represented 37% more labour hours lost for this reason than annually in the period from 1990–1999 [5].

In Australia, outdoor workers are routinely exposed to environmental conditions conducive to heat stress [6]. At warmer air temperatures, heat flux from a body to the environment decreases (and can reverse when the air temperature is above body temperature). The evaporation of sweat takes up latent heat to cool a body, but the effect is reduced at higher air humidities because the air has less capacity to take on moisture. Exposure to environments with both high temperatures and humidities, therefore, limits body heat dissipation. Moreover, protective clothing creates an insulative barrier between the environment and skin, further reducing body heat loss. Exacerbating these reductions in heat loss, the physical effort of work elevates metabolic heat production and radiant heat emanating from surrounding structures and materials increase heat gain; meaning more heat must be shed to maintain a healthy body temperature. Where body heat loss cannot match the rate of internal heat production or gain, uncompensable heat stress occurs, which requires risk management strategies to avoid excessive elevations in workers’ heat strain [7].

Increasing heat exposure for outdoor workers due to climate change in Australia has been reported [8]. Hall et al. [8] projected future potential wet-bulb globe temperatures (WBGT) using modelled changes in temperature and vapour pressure from a representative global climate model. Four commonly applied potential future climate scenarios were applied, which comprised two representative concentration pathways (RCPs), RCP4.5 and RCP8.5, over two time periods, 2041–2060 and 2061–2080. Environmental conditions were predicted across several seasons, times of day, and wind-speed scenarios. Maps of future daytime summer WBGT under all projected climate scenarios indicated significantly larger geographic areas subject to higher heat categories commonly used to determine physical work limits. Consequently, outdoor workers exposed to these projected future conditions may be at increased risk of elevated heat strain and heat-related illness.

For outdoor workers in conditions of high heat stress, the World Health Organization (WHO) and International Organisation for Standardisation (ISO) indicate that “it is inadvisable for deep body temperature to exceed 38.0 °C in prolonged daily exposure to heavy work” [9,10]. Moreover, to avoid the deleterious effects of dehydration, the ISO recommends that body mass loss due to sweating should not exceed 5% of body mass [10]. Using these workplace heat strain limits and the projected environmental conditions, we can now evaluate the likely impact of climate change on the heat strain and labour capacity of outdoor workers in Australia. Therefore, this study’s aim was to compare predicted worker heat strain (core temperature elevation and sweat loss) and labour capacity in current and future environmental conditions across various locations in Australia. Linking predicted WBGT forecasts to heat strain will inform planning to develop near-future workplace heat risk mitigation strategies.

## 2. Methods

### 2.1. Environments

Scenario 1 of Hall et al. [8] was considered for the purposes of this study, representing the average summer temperature and humidity in January (the warmest month of the year), with a clear atmosphere at 3 pm and a low wind speed (0.5 m/s). The climate data for the following locations were included in the current modelling: Darwin, Townsville, Tom Price, Griffith, Port Macquarie, and Clare (Table 1). The locations span some of the hottest northern regions of Australia and include coastal and inland areas of the eastern and south-eastern regions (Figure 1).

### 2.2. Predicted Heat Strain

Thermophysiological responses to the ambient conditions listed in Table 1 were estimated using the Predicted Heat Strain (PHS) model (ISO7933:2004 [10]), implemented through the software provided and validated by the FAME laboratory [11]. The PHS model predicts safe work durations before threshold values of rectal (core) temperature and sweat loss are attained. Specifically, to reduce the risk of any worker reaching dangerously high body temperatures, work duration limits based on core temperature elevation aim to ensure work is ceased when the core temperature reaches 38.0 °C, on average. Similarly, to protect 95% of workers from the deleterious effects of dehydration, total sweat loss is limited to 5% of body mass. The modelling assumed a 60% fluid replenishment rate, i.e., 60% of the body fluid lost through sweat loss was replenished through fluid ingestion during the period of work. Although fluid consumption rates vary considerably, this was selected to generally reflect outdoor workers [12], and it coincides with the ad libitum assumption of the model, that workers can drink freely [10,11].

Safe work durations were evaluated in each of the locations and climate scenarios (Table 1), accounting for the effects of metabolic rate (indicating workloads), work clothing, and individual characteristics. Light (115 W/m^2^), moderate (145 W/m^2^), and heavy (200 W/m^2^) metabolic rates, as defined by ISO7933 [10], were included in the modelling. Mechanical efficiency was predicted by the Fiala method as described by Ioannou et al. [11]. Intrinsic clothing insulation was assumed to be 0.9 clo, representing standard workwear clothing ensemble consisting of underpants, long trousers, a long-sleeved shirt, calf-length socks, and boots, based on similar ensembles outlined in ISO9920 (see Table C3, ensemble numbers 10 and 12, in reference [13]). Workers were assumed to be heat acclimatised and consuming water ad libitum to replace 60% of sweat loss [12]. Other model parameters included body height (assumed to be 180 cm), body mass (85 kg), and posture (standing). The maximum model duration was 300 min (5 h), as this is the maximum duration of continuous work without a break permitted by government regulations in Australia.

### 2.3. Analysis

Predicted heat strain modelling was performed using the FAME laboratory’s Predicted Heat Strain modelling software (version 1.0) (FAME Laboratory, University of Thessaly, Karies, Greece). Data analysis and visualisation were performed using R studio (version 1.3.1093)(Posit PBC, Boston, MA, USA) with the following packages: base [14], rio [15], stringr [16], dplyr [17], ggplot2 [18], and lemon [19].

## 3. Results

The modelled core temperature elevation during light, moderate, and heavy work varied considerably by location and climate scenario (Figure 2). In Clare, the modelled core temperature elevation stabilised below 38 °C during each work intensity in both baseline conditions and each predicted climate scenario. Similarly, in Griffith, the modelled core temperature elevation did not exceed 38 °C in the baseline conditions or RCP4.5 climate scenarios; however, the modelled core temperature elevation rose above 38 °C in all metabolic rate categories in the RCP8.5 scenarios. In Port Macquarie, the modelled core temperature elevation did not exceed 38 °C in the baseline conditions. However, with the warming conditions in the RCP4.5 and RCP8.5 climate scenarios, a progressive elevation in modelled core temperature that surpassed 38 °C was observed (except during light work in the RCP4.5 2041–2060 scenario). For Darwin, Townsville, and Tom Price, excessive elevations in modelled core temperature were predicted even in baseline conditions, surpassing 38 °C. Concerningly, the rate of core temperature elevation in these locations increased progressively with the warming climate scenarios.

Consequent to the varied rates of modelled core temperature elevation, the predicted time to attain a core temperature of 38 °C (safe work duration) varied with location, climate scenario, and metabolic rate (Table 2). The greatest relative decreases in safe work durations associated with predicted warming climate conditions were observed in locations where the core temperature elevations were not predicted to exceed 38 °C in baseline conditions—Port Macquarie (both RCP 4.5 and 8.5 scenarios) and Griffith (only RCP 8.5 scenarios) (Table 2). This was particularly the case for the RCP8.5 scenarios, where the relative safe work durations for work of any intensity were predicted to fall below 50% of the baseline durations in both time periods (2041–2060 and 2061–2080) for Port Macquarie and for the 2061–2080 period in Griffith. Safe work durations at all work intensities were also predicted to be progressively lower with warming climate conditions in Tom Price, Darwin, and Townsville (Table 2); however, in these locations, the relative work durations ranged between ~50 and 90% of the baseline predictions for the RCP4.5 and RCP8.5 model predictions (Table 2).

In the majority of scenarios, predicted heat strain indicated that workers were likely to reach the safe core temperature limit before the safe sweat loss limit. Interesting, however, is the observation that in locations where the predicted core temperature either did not reach the safe limit or allowed prolonged work durations before being attained, reaching the sweat loss limit was more likely to necessitate restricting work duration (Table 3). This was the case in Clare, as 5% body mass loss due to sweating occurred in less than 300 min in all heavy work scenarios, all moderate work scenarios (except in the baseline climate scenario), and even during light work in the RCP8.5 2061–2080 climate scenario. Similarly, in Port Macquarie in the baseline climate scenario and for light work in the RCP 4.5 2014–2060 scenario, and in Griffith in all baseline, RCP 4.5, and RCP 8.5 climate scenarios for light and moderate work, the sweating-related body mass loss limit was reached within 300 min of work.

## 4. Discussion

The present study demonstrates that projected future climatic conditions in summer will adversely impact the heat strain and labour capacity of outdoor workers across Australia. In the northern regions (Tom Price, Darwin, and Townsville), where outdoor workers are already subjected to restricted work durations due to the risk of excessive elevations in core temperature, further restrictions will have to be imposed as global heating progresses. In two of the southern regions (Griffith and Port Macquarie), the warming climate will expose workers to conditions requiring work duration limits to prevent excessive core temperature elevations. In both time periods considered in the RCP8.5 scenarios, these limits will restrict labour capacity in Port Macquarie to less than 50% of those calculated for the baseline climate, 1986–2005, and this will also be the case in the 2061–2080 time period in Griffith. Furthermore, even in locations such as Clare, where modelled warming climatic conditions would not expose outdoor workers to excessive elevations in core temperature, greater sweat rates will be required to ensure adequate evaporative heat loss, with these sweat rates predisposing workers to the risk of dehydration. Overall, to adequately protect outdoor workers from the deleterious effects of heat, risk management strategies will need to adapt to the warming climatic conditions expected during Australian summers.

### 4.1. Heat Strain and Labour Capacity

Currently, in many regions across Australia, environmental conditions are conducive to heat stress, with wet-bulb globe temperatures commonly in excess of 25 °C or even 30 °C [20,21,22,23,24,25]. These levels of heat stress require careful management of work duration, as recommended by the ISO [26], as work in such conditions places strain on the physiological systems of the body that regulate body temperature. Heat strain during work is evidenced by an elevated heart rate, sweat rate, and core temperature [20,21,22,23,24,25]. Excessive core temperature elevation predisposes workers to heat exhaustion or, in severe cases, heat stroke [27,28].

Australian workers commonly (>90%) report being affected by heat on hot days [29], with fatigue, irritability, headache, dizziness, fainting, and nausea the most commonly reported symptoms [29,30], so much so that they have been classified as chronic [31]. Moreover, symptoms of exertional heat illness are common among outdoor workers in northern Australia, particularly in the mining and construction industries and in the military and first responders [20,24,27,32,33]. Whilst experiencing heat stress symptoms may not indicate medically reportable cases of heat-related illness, it does suggest that the physiological systems of the body may be struggling to meet the demands of thermoregulation. Indeed, moderate to high levels of heat strain are observed in Australian workplaces [23,27]. Although data on medically reported cases of exertional heat illness are scarce, heat exhaustion has been reported in the mining industry, with dehydration and the summer period leading to the greatest risk [28]. Overall, managing occupational heat stress in Australia is a seasonal challenge, and under the climate scenarios considered in this study, could pass thresholds into greater heat strain, more frequent symptomology, and cases of heat exhaustion or, worse, heat stroke.

With the intention of mitigating the risk of excessive heat strain, the WHO and ISO recommend limiting work duration to minimise workers becoming exhausted and to prevent excessive elevations in core temperature that may lead to heat stroke [9,10]. However, limitations to work duration also reduce labour capacity and worker productivity. In agreement with findings of a recent literature review examining the impact of environmental conditions on worker heat strain and labour capacity [1,2] and consistent with observed global reductions in labour capacity of 37% since 1990–1999 [5], our modelling is evidence that global heating will further exacerbate heat strain and impair Australian labour capacity in the absence of effective management strategies. Of particular concern are the most notable increases in predicted heat strain and associated reductions in labour capacity that will likely occur in regions of Australia not traditionally exposed to high heat stress—specifically demonstrated for Griffith and Port Macquarie. While increasing work duration limitations were predicted by our modelling to be required in the already-high heat-stress contexts of the northern regions as climate change progresses, greater relative reductions in labour capacity (or safe work durations) were predicted for the southern regions (Figure 2 and Table 2).

A critical mechanism underpinning the human thermal balance is the evaporation of sweat. For the core temperature to stabilise, adequate fluid replenishment is required to prevent dehydration. The present modelling shows that, in locations where core temperature can stabilise at safe levels during work at some or all intensities in some climate scenarios (Clare, Griffith, and Port Macquarie), it does so through higher sweat rates and fluid losses (Table 3). Therefore, if fluid replenishment practices are left unchanged, workers will reach 5% body mass loss progressively earlier with warming climatic conditions. Dehydration predisposes workers to an elevated core temperature, cardiovascular strain, and risk of heat exhaustion [28]. Historically, adequate fluid replenishment has been an effective strategy to mitigate the risk of dehydration in Australian workers [6,22,34]. However, the projected environmental conditions will lead to increased fluid requirements during work in the heat, particularly around hydration monitoring and replenishment strategies.

### 4.2. Risk Management

The development of heat health action plans at national, local, and institutional levels will be critical in preparing for climate change [35]. Various organisations provide resources for work in the heat. The ISO’s series on Ergonomics of the Thermal Environment provides comprehensive guidance on managing the risks of hot workplaces. Similarly, the Australian Institute of Occupational Hygienists’s *A Guide to Managing Heat Stress* provides practical guidance for the Australian environment [36]. Broadly, risk management involves tools and techniques for assessing the level of heat stress to which workers are exposed and implementing strategies to mitigate the associated risks.

Monitoring daily environmental conditions is imperative for heat stress management, as these conditions determine the maximum potential for body heat dissipation. As a testament to their importance, 340 heat stress indices have been developed [37] to determine the risk of heat stress and inform the selection of appropriate risk management strategies. Of these indices, WBGT is the most utilised globally [38] and was recently acknowledged as having the highest potential to assess the physiological strain experienced by individuals working in the heat [39]. While the WBGT index is broadly applicable across a wide range of conditions, an important limitation should be acknowledged regarding conditions with high humidity. In the current modelling, the predicted heat strain was higher in Port Macquarie compared to Griffith in spite of slightly lower WBGT predictions. These differences can be attributed to the very high humidity in Port Macquarie, conditions in which WBGT is known to underestimate the potential for heat dissipation [40,41,42].

A range of strategies can be implemented to mitigate heat stress in the workplace, including hydration practices, work–rest scheduling, adjusting work intensity, and cooling strategies. The selection of appropriate strategies for a given workplace needs to consider the environmental conditions, work rate, protective clothing, and facilities and resources available and should be informed by evidence of their effectiveness. To alleviate heat stress, fluid consumption is reported as the primary strategy used by almost 90% of Australian workers exposed to heat stress, often in combination with rest (44%) or cooling strategies (67%) [30]. Replacing fluid lost in sweat helps to mitigate the detrimental effects of dehydration, which progressively increases physiological strain and perceived exertion [43]. While considerable debate continues regarding optimal drinking strategies (e.g., drinking to thirst or scheduled drinking), broad guidance recommends limiting dehydration to 2% or 5% of body mass loss in athletic and occupational settings, respectively [10,44]. However, workers should also be cautioned against the overconsumption of hypotonic fluids (drinking volumes of water in excess of sweat losses), which can lead to hyponatremia and, if this is severe, to serious illness or death due to cardiovascular and neurological complications.

While maintaining adequate hydration is important, workers should be aware that good hydration alone will not protect them from excessive elevations in core temperature when work rates and internal heat production exceed the rate at which body heat can be lost. As shown in the present modelling, the core temperature limit was often achieved before the sweat-related body mass loss limit (Table 3). In such conditions, rest periods and adjustments to work–rest cycles and cooling strategies need to be implemented alongside hydration practices.

While light-intensity work and the ability to self-pace tasks have spared many outdoor workers from excessive heat strain in the past [6], this strategy will become more challenging in the future. The current modelling shows that maximum safe durations of even light-intensity work will be reduced by as much as 20%–50% in Tom Price, Darwin, Townsville, and as much as 60–70% in Griffith and Port Macquarie as climate changes progress (Table 2). Similar relative reductions in labour capacity can be expected for heavy work, characteristic of military training exercises and some other roles. If the predicted future climate scenarios develop, rescheduling work to cooler parts of the day or allowing longer rest periods may be required.

Resting in the shade is often recommended and utilised; however, the effectiveness of this strategy is affected by environmental conditions and barriers to body heat loss, such as worker attire. Core temperature cooling rates of just 0.2 °C per 10 min result from rest periods while wearing industrial, military, or firefighting personal protective equipment despite WBGTs of less than 28.0 °C [45]. While removing protective attire improves cooling rates, passive rest generally produces inferior results to active cooling methods.

Finally, technological advancements may also lead to improving heat stress management strategies. The mechanisation of work tasks may alleviate the physical demands of, and thereby metabolic rates associated with, certain tasks, which may reduce workers’ heat strain [46].

### 4.3. Assumptions and Limitations

A range of assumptions should be considered when interpreting the outcomes of the present study. First, the predictions were based on an average day in January (summer), at 3 pm, with low wind speed (0.5 m/s). As Hall et al. [8] show, conditions vary seasonally and over the course of a day. Moreover, this modelling does not account for heat events (heat waves), which may change day-to-day conditions considerably. Therefore, it is always important to monitor the conditions of the day and manage risks accordingly. Second, the present heat strain modelling only considered workers wearing basic protective clothing, including underwear, long-sleeve shirts, and trousers. Additional clothing items for protection against specific hazards should be considered in assessing workplace risks for heat stress, as they affect the rate of change in core temperature. In contrast, wearing short-sleeved shirts and shorts may reduce the predicted heat strain (depending on environmental conditions and work intensity) [47], but caution is advised regarding the risk of sunburn to uncovered skin and melanoma. Third, tolerance for working in the heat will vary between individuals due to a range of factors such as their health and fitness, age, experience, acclimatisation status, and hydration practices. Notwithstanding these assumptions, this heat strain modelling provides insight into the likely effects climate will have on workers’ heat strain. Further modelling is warranted to elucidate the likely impacts of these and other factors.

## 5. Conclusions

In conclusion, the predicted heat strain of outdoor workers in Australia during light, moderate, and heavy work varied considerably by location and climate scenario. As climate change progresses, our modelling shows that even more restrictive work duration limits may be required in the already-high heat-stress contexts of northern Australia, and that greater relative reductions in labour capacity were predicted for the south-eastern regions. To adequately protect outdoor workers from the deleterious effects of heat, risk management strategies will need to adapt to the warming climatic conditions expected during summer across Australia.

## Figures and Tables

**Figure 1 ijerph-20-05675-f001:**
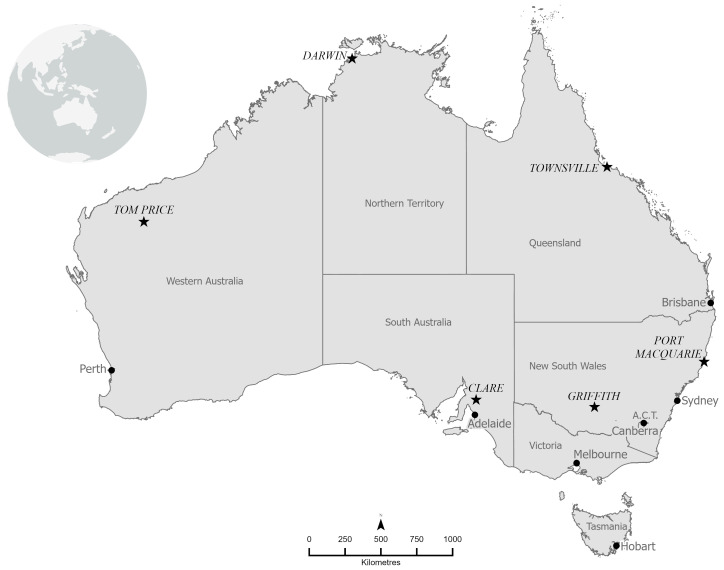
Map of Australia with the cities included in the current modelling (stars) and capital cities (dots). Map created by the Spatial Data Analysis Network (SPAN), Charles Sturt University 2023.

**Figure 2 ijerph-20-05675-f002:**
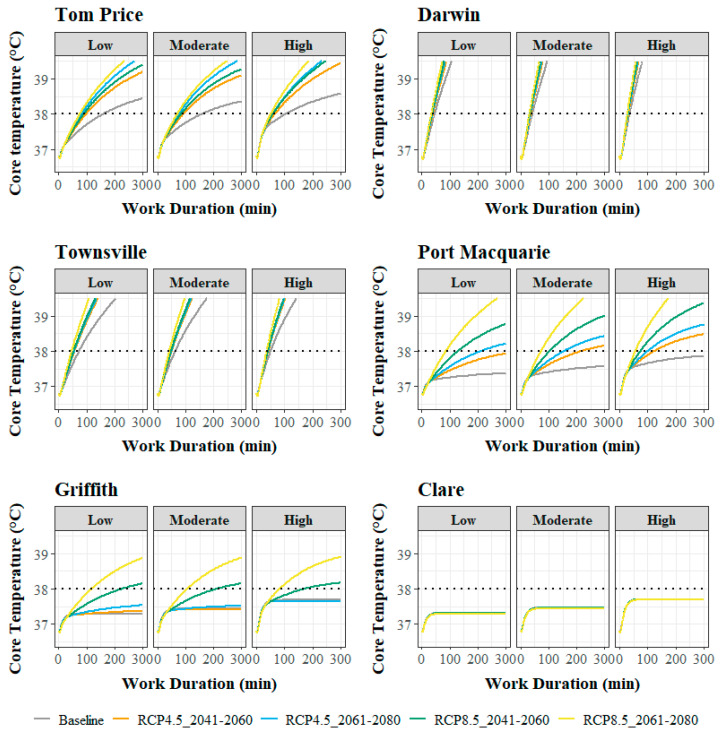
Modelled core temperature elevation for each location, work intensity (low, moderate, and high) and climate scenario. The dashed horizontal line indicates the core temperature limit of 38.0 °C.

**Table 1 ijerph-20-05675-t001:** Baseline and forecast environmental conditions on an average day in January (summer) at 3 pm with low wind (0.5 m/s) across the study locations.

		Darwin	Tom Price	Townsville	Griffith	Clare	Port Macquarie
BASELINE	Ta (°C)	32.6	38.9	31.5	32.2	29.3	26.7
1986–2005	Tg (°C)	47.4	53.3	45.6	46.9	44.2	40.9
	Tnwb (°C)	29.5	24.8	27.6	22.8	20.6	24.6
	RH (%)	79.5	30.9	74.1	44.2	44.8	84.2
	WBGT (°C)	33.4	31.9	31.6	28.6	26.2	28
RCP4.5	Ta (°C)	33.6	41.0	32.4	33.1	30.9	27.1
2041–2060	Tg (°C)	48.4	55.3	46.6	47.9	45.8	41.3
	Tnwb (°C)	30.4	25.6	28.6	23.8	21.6	25.3
	RH (%)	79.2	28.4	75.1	45.5	43.4	86.5
	WBGT (°C)	34.4	33.1	32.6	29.6	27.4	28.6
RCP4.5	Ta (°C)	34.0	41.2	32.7	33.9	31.4	27.9
2061–2080	Tg (°C)	48.8	55.5	46.9	48.6	46.3	42.1
	Tnwb (°C)	30.7	26.1	28.7	24.0	21.7	25.6
	RH (%)	78.7	29.6	74.0	43.4	41.9	83.1
	WBGT (°C)	34.7	33.5	32.8	29.9	27.6	29.0
RCP8.5	Ta (°C)	34.6	42.6	33.0	34.6	31.6	28.5
2041–2060	Tg (°C)	49.4	56.6	47.2	49.4	46.5	42.8
	Tnwb (°C)	30.9	25.8	28.8	24.7	21.8	26.2
	RH (%)	76.6	25.4	72.9	44.1	41.6	83.3
	WBGT (°C)	35.0	33.6	32.9	30.7	27.8	29.7
RCP8.5	Ta (°C)	35.2	43.7	33.6	36.3	33.2	29.8
2061–2080	Tg (°C)	50.0	57.6	47.9	51.0	48.0	44.1
	Tnwb (°C)	31.4	26.3	29.5	25.5	22.7	27.1
	RH (%)	76.2	24.6	73.8	41.6	39.9	81.0
	WBGT (°C)	35.5	34.3	33.6	31.7	28.9	30.7

Data from Hall et al. [8]. Ta: Air temperature; Tg: Globe temperature; Tnwb: Natural wet bulb temperature; RH: Relative humidity; WBGT: Wet-bulb globe temperature. Four commonly applied potential future climate scenarios were applied, which comprised two representative concentration pathways (RCPs), RCP4.5 and RCP8.5, over two time periods, 2041–2060 and 2061–2080.

**Table 2 ijerph-20-05675-t002:** Predicted time (min (percent of time associated with baseline climate scenario)) to attain a core temperature of 38.0 °C for each location, climate scenario, and work intensity.

Location	Work	Baseline	RCP4.5 2041–2060	RCP4.5 2061–2080	RCP8.5 2041–2060	RCP8.5 2061–2080
CLARE	Light	NL (100)	NL (100)	NL (100)	NL (100)	*NL (100)*
	Moderate	NL (100)	*NL (100)*	*NL (100)*	*NL (100)*	*NL (100)*
	Heavy	*NL (100)*	*NL (100)*	*NL (100)*	*NL (100)*	*NL (100)*
PORT MACQUARIE	Light	*NL (100)*	*NL (100)*	211 (70)	129 (43)	86 (29)
	Moderate	*NL (100)*	216 (72)	154 (51)	103 (34)	72 (24)
	Heavy	*NL (100)*	122 (41)	98 (33)	73 (24)	56 (19)
DARWIN	Light	46 (100)	40 (87)	38 (83)	37 (80)	35 (76)
	Moderate	42 (100)	36 (86)	35 (83)	34 (81)	32 (76)
	Heavy	36 (100)	32 (89)	30 (83)	30 (83)	28 (78)
TOM PRICE	Light	160 (100)	98 (61)	83 (52)	90 (56)	76 (48)
	Moderate	159 (100)	92 (58)	76 (48)	84 (53)	70 (44)
	Heavy	104 (100)	65 (63)	58 (56)	58 (56)	51 (49)
TOWNSVILLE	Light	73 (100)	57 (78)	55 (75)	54 (74)	47 (64)
	Moderate	63 (100)	50 (79)	49 (78)	48 (76)	42 (67)
	Heavy	50 (100)	42 (84)	41 (82)	40 (80)	36 (72)
GRIFFITH	Light	*NL (100)*	*NL (100)*	*NL (100)*	*229 (76)*	118 (39)
	Moderate	*NL (100)*	*NL (100)*	*NL (100)*	*214 (71)*	106 (35)
	Heavy	*NL (100)*	*NL (100)*	*NL (100)*	174 (58)	83 (28)

NL—no limit (core temperature was not predicted to rise above 38.0 °C within the maximum modelled work duration of 300 min). Values in *italics* indicate the sweat loss limit was reached within 300 min. Where baseline climate scenarios were associated with no heat stress-induced work duration limit, the maximum potential work duration of 300 min was used to calculate the relative reduction from the baseline climate scenario in work duration for each of the projected climate scenarios.

**Table 3 ijerph-20-05675-t003:** Predicted time (min) to attain a sweat loss of 5% body mass (assuming a 60% fluid replenishment rate) for each location, climate scenario, and work intensity.

Location	Work	Baseline	RCP4.5 2041–2060	RCP4.5 2061–2080	RCP8.5 2041–2060	RCP8.5 2061–2080
CLARE	Light	NL	NL	NL	NL	269
	Moderate	NL	284	278	274	245
	Heavy	267	242	238	235	214
PORT MACQUARIE	Light	299	271	*252*	*220*	*185*
	Moderate	260	*236*	*221*	*196*	*179*
	Heavy	216	*197*	*187*	*180*	*178*
DARWIN	Light	*177*	*177*	*177*	*177*	*177*
	Moderate	*176*	*176*	*176*	*176*	*176*
	Heavy	*176*	*176*	*176*	*176*	*176*
TOM PRICE	Light	*185*	*176*	*176*	*176*	*176*
	Moderate	*178*	*176*	*176*	*176*	*175*
	Heavy	*177*	*175*	*175*	*175*	*174*
TOWNSVILLE	Light	*178*	*176*	*176*	*176*	*117*
	Moderate	*177*	*176*	*176*	*176*	*176*
	Heavy	*177*	*176*	*176*	*176*	*175*
GRIFFITH	Light	280	242	232	210	*184*
	Moderate	253	220	213	194	*178*
	Heavy	219	195	188	*179*	*177*

Values in *italics* indicate that the core temperature limit of 38.0 °C was predicted to be attained (and work therefore stopped) before the sweat loss limit was reached. NL—no limit (sweat loss was not predicted to exceed 5% of body mass during the maximum modelled work duration of 300 min).

## Data Availability

The data presented in this study are available upon reasonable request from the corresponding author.

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
