# Peer review of "Climate Change Effects on the Predicted Heat Strain and Labour Capacity of Outdoor Workers in Australia"

_ijerph, 2023, doi:10.3390/ijerph20095675_

Round 1

Reviewer 1 Report

General Comments:

Using the Predicted Heat Strain (PHS) model, the present study compared expected worker heat strain (core temperature elevation and sweat loss) and labour capacity under current and future environmental conditions in several Australian locations (ISO7933:2004). The results of the study indicate that expected future climatic conditions in Australia's summer will negatively effect labour capability among outdoor workers. In the absence of adequate management techniques, global warming would worsen heat stress and reduce Australia's labour capacity, according to the study's modelling. This correlation between anticipated WBGT and heat stress will inform the development of near-future workplace heat risk mitigation methods. The manuscript could be suitable for publication if the following suggestions are implemented.

ABSTRACT

1.     Total word count exceeds 200.

2.     Line 16: Mention the method used to calculate/estimate the WBGT.

 HIGHLIGHTS

3.     Line 35: Reduction in Labour capacity in northern Australia is mentioned as 50%–80% from the baseline in the abstract. This result is not consistent with the highlights.

1.     Line 37: “Risk management strategies must adapt to warming conditions to protect outdoor workers from the deleterious effects of heat” – the same line is mentioned above in the abstract in the 27th line. Kindly rephrase the sentence in any one section.

INTRODUCTION

 The introduction is well written with few minor corrections

2.     Line 64: “Increasing heat exposure for outdoor workers due to climate change in Australia has been reported”- cite this sentence with the latest citation.

METHODOLOGY:

 1.     Line 90: replace the word “summertime” with summer time.

2.     Figure 1: In the title change the word capitol to capital.

3.     Line 90: State why only January month was considered for the environmental data.

4.     Table 1: Mention the source from which the data was taken.

5.     Line 127: Were the age and sex considered in the model for the prediction of CBT?

6.     Line 126: Replace the word “ad” with “and”.

7.     Line 132: This sentence must be moved to section 2.2

 RESULTS

 The results section is also well written with few comments

8.     Lines 144-146: "Contrarily, in both the RCP4.5 and RCP8.5 climatic scenarios, a gradual increase in core temperature that exceeded 38 °C was seen (except during light work in the RCP4.5 2041-2060 scenario)." Give possible explanations for this.

9.     Highlight the reason for differences between cities for Sweat loss and CBT rise.

10.  Interesting prediction, but will nobody be working in Townsville, Tom Price, and Darwin as even at baseline CBT has reached 38°C? If that is the case, no one can work in other tropical settings in summer. What message are you trying to convey? Be clear.   

DISCUSSION

The discussion section is also well written with few comments

·       Line 217: Cite some latest references for this sentence “Heat strain during work is evidenced by elevated heart rate, sweat rate, and core temperature. Excessive core temperature elevation predisposes workers to heat exhaustion or in severe cases, heat stroke.”

·       Line 285 and 286: “A range of strategies can be implemented to mitigate heat stress in the workplace, including hydration practices, work-rest scheduling, adjusting work intensity, and cooling strategies”. Specify which technique is more effective and at what interval the break should be taken, or whether it should be dependent on their working hours.

 Conclusion: The author has concluded the study very well in a comprehensive manner but can still add a few points on risk management strategies based on your study results.

Author Response

The authors thank the reviewer for their thoughtful and thorough review of our manuscript. We have addressed each point raised by the reviewer and believe the manuscript has improved as a result. Specifically:

Abstract

Line 16: we have reworded this sentence to better reflect the methods regarding the environmental data used in this study. The word count of the abstract exceeds 200 but is in accordance with the journal’s author instructions of being “about 200 words maximum.”

Highlights

Line 35: Thank you for highlighting this point. The two phrasings were consistent, as 80% of baseline is equal to a 20% reduction. However, to avoid this confusion, we have rephrased the abstract to consistently describe the reduction in labour capacity.

Line 37: The third highlight has been rephrased.

Introduction

Line 64: a reference has now been included at the end of this sentence.

Methodology

Line 90: spelling corrected.

Figure 1: spelling corrected.

Line 90: we have now indicated that January was chosen as the warmest month of the year. This is when the risk of heat stress is greatest and when risk management is critical.

Table 1: The source of the data has now been cited in the footnotes.

Line 126: The term “ad libitum” is spelt correctly.

Line 127: no, the model does not include age or sex as input variables. Individuals’ characteristics has been noted in the limitations in the discussion.

Line 132: the following sentence is also included in section 2.2: “Thermophysiological responses to the ambient conditions listed in Table 1 were estimated using the Predicted Heat Strain (PHS) model (ISO7933:2004 (11)), implemented through the software provided and validated by the FAME laboratory (12).”

Results

Line 144-146: further detail has been added to this sentence to indicate it is the warming conditions that are responsible for the observed change in core temperature response.

Point 9: The reasons for differences between locations for CBT rise and sweat loss is captured in lines 177-180, “Interesting, however, is the observation that in locations where predicted core temperature either did not reach the safe limit or allowed prolonged work durations before being at-tained, reaching the sweat loss limit was more likely to necessitate restricting work duration.”

Point 10: The modelling does not show that work cannot be performed in any location at any time, but it does show that the duration of work that can be performed becomes progressively more restricted as conditions warm, due to reaching the core temperature threshold more quickly.

Discussion

Line 217: citations have been added to these sentences.

Line 285: the following sentence has been included to highlight important considerations when selecting the risk management strategies, “Selection of appropriate strategies for a given workplace needs to consider the environmental conditions, work rate, protective clothing, and facilities and resources available.”

Conclusion

While risk management strategies are described in the discussion, our study itself has not evaluated specific risk management strategies. As such, we have not drawn conclusions about which risk management strategies should be implemented, as this is highly dependent on the context of each workplace.

Reviewer 2 Report

General Comments

This is an excellently written manuscript describing the risk of exceeding core temperature and hydration thresholds in workers in hot environments across Australia.  The authors are commended for their excellent idea, clear writing, and attention to detail.

Specific Comments

Ln 114: Please explain why a 60% fluid replacement rate was chosen for the model here.  You include a citation to support this assumption later, but here would be appropriate as well since it’s the first time it’s mentioned.  Is there a more recent citation that also identifies ad libitum water consumption to reflect modern workers?

Figure 2: Color blind individuals will not be able to discern some of the colors in this figure. Consider adjusting the line patterns to show differences among models.  A dashed horizontal line indicating the 38C would be beneficial.

Table 3. I am struggling with the idea of not including the 5% BML data for many cities and intensities because they would have reached the 38C cut-off.  The same could be said in the opposite way, that some Table 2 values should not be included because they would have already reached the 5% BML cut-off, but this was not presented. Either all data should be included for all cities/intensities (preferred), or Table 2 data should not be included and replaced with “.” when the 5% cut-off would have been reached.

Author Response

The authors thank the reviewer for their thoughtful and thorough review of our manuscript. We have addressed each point raised by the reviewer and believe the manuscript has improved as a result. Specifically:

Line 114: The reference for the chosen fluid consumption rate has been brought forward and a rationale has been provided in the text. The authors have searched the literature but there is scarce information available on fluid consumptions rates among outdoor workers. The chosen fluid replacement rate also coincides with the ad libitum approach of the model, assuming that workers are able to drink freely.

Figure 2: Thank you for highlighting these points. We have adjusted the line colour palette to ensure it is suitable for readers who are colourblind. We have also included a horizontal dashed line at 38°C.

Table 3: Thank you for highlighting this point. We have now included all data in both Tables 2 and 3. When a limit is earlier in the other table, the value has been presented in italics.